# Ablation of Sphingosine Kinase 1 Protects Cornea from Neovascularization in a Mouse Corneal Injury Model

**DOI:** 10.3390/cells11182914

**Published:** 2022-09-17

**Authors:** Joseph L. Wilkerson, Sandip K. Basu, Megan A. Stiles, Amanda Prislovsky, Richard C. Grambergs, Sarah E. Nicholas, Dimitrios Karamichos, Jeremy C. Allegood, Richard L. Proia, Nawajes Mandal

**Affiliations:** 1Dean A. McGee Eye Institute, University of Oklahoma Health Sciences Center, Oklahoma City, OK 73104, USA; 2Department of Nutrition and Integrative Physiology, University of Utah, Salt Lake City, UT 84112, USA; 3Department of Ophthalmology, Hamilton Eye Institute, University of Tennessee Health Sciences Center, Memphis, TN 38163, USA; 4North Texas Eye Research Institute, University of North Texas Health Science Center, Fort Worth, TX 76107, USA; 5Department of Pharmaceutical Sciences, University of North Texas Health Science Center, Fort Worth, TX 76107, USA; 6Department of Pharmacology and Neuroscience, University of North Texas Health Science Center, Fort Worth, TX 76107, USA; 7Department of Biochemistry and Molecular Biology, Virginia Commonwealth University School of Medicine, Richmond, VA 23298, USA; 8Genetics of Development and Disease Branch, National Institute of Diabetes and Digestive and Kidney Diseases, National Institutes of Health, Bethesda, MD 20892, USA; 9Departments of Anatomy and Neurobiology, and Pharmaceutical Sciences, University of Tennessee Health Sciences Center, Memphis, TN 38163, USA; 10Memphis VA Medical Center, Memphis, TN 38104, USA

**Keywords:** cornea, neovascularization, sphingolipid, sphingosine-1-phosphate, sphingosine kinase 1

## Abstract

The purpose of this study was to investigate the role of sphingosine kinase 1 (SphK1), which generates sphingosine-1-phosphate (S1P), in corneal neovascularization (NV)**.** Wild-type (WT) and *Sphk1* knockout (*Sphk1*^−*/*−^) mice received corneal alkali-burn treatment to induce corneal NV by placing a 2 mm round piece of Whatman No. 1 filter paper soaked in 1N NaOH on the center of the cornea for 20 s. Corneal sphingolipid species were extracted and identified using liquid chromatography/mass spectrometry (LC/MS). The total number of tip cells and those positive for ethynyl deoxy uridine (EdU) were quantified. Immunocytochemistry was done to examine whether pericytes were present on newly forming blood vessels. Cytokine signaling and angiogenic markers were compared between the two groups using multiplex assays. Data were analyzed using appropriate statistical tests. Here, we show that ablation of SphK1 can significantly reduce NV invasion in the cornea following injury. Corneal sphingolipid analysis showed that total levels of ceramides, monohexosyl ceramides (HexCer), and sphingomyelin were significantly elevated in *Sphk*^−*/*−^ corneas compared to WT corneas, with a comparable level of sphingosine among the two genotypes. The numbers of total and proliferating endothelial tip cells were also lower in the *Sphk1*^−*/*−^ corneas following injury. This study underscores the role of S1P in post-injury corneal NV and raises further questions about the roles played by ceramide, HexCer, and sphingomyelin in regulating corneal NV. Further studies are needed to unravel the role played by bioactive sphingolipids in maintenance of corneal transparency and clear vision.

## 1. Introduction

The cornea is a dome-shaped tissue covering the anterior-most portion of the eye. It is responsible for both protection and refraction of light in the eye [1]. The tissue must remain clear for light to pass through the eye unhindered, which is essential for clear vision. Corneal transparency is maintained through an intricate arrangement of collagen in the stroma layer of the cornea [2]. Transparency is also attributed to the absence of blood vessels in the cornea. To maintain this angiogenic privilege, the cornea has evolved vascular endothelial growth factor (VEGF)-A traps that keeps this angiogenic factor in check [3]. The balance of angiogenic and growth factors can be offset by infection, injury, or disease, which can trigger neovascularization (NV) into the stroma of the cornea. As this occurs, there is an increase in scarring, lipid disposition, and edema that can render opacity of the cornea [4]. Corneal opacity resulting from NV affects an average of 1.4 million people per year; it can lead to blindness and is the primary cause of failed corneal transplants [4,5,6]. Another important contributor of corneal opacity is fibrosis resulting from an injury. Corneal fibrosis is a leading cause of blindness worldwide and is caused by pathological overaccumulation of disorganized extracellular matrix (ECM) produced by activated fibroblasts and myofibroblasts in the corneal stroma [7]. However, the molecular mechanisms underlying corneal NV and fibrosis is poorly understood.

The idea that sphingolipids, a subgroup of cellular lipids, are part of cellular signaling networks in addition to their role as a structural component in cells is now widely accepted. Research from the last two decades has established several sphingolipids as signaling molecules involved in multiple cellular functions ranging from apoptosis, cell survival, proliferation, and differentiation [8,9,10,11,12,13,14,15,16]. The central player in cellular sphingolipid homeostasis is ceramide, which is also one of the major bioactive sphingolipids [17]. Ceramide can be synthesized via the de novo pathway, the hydrolysis of phosphorylcholine from sphingomyelin, or the recycling of sphingosine [18,19,20]. While ceramide is a required lipid species in the plasma membrane, it can also play a critical role in inducing apoptosis within the cell [21,22]. Another important bioactive sphingolipid is sphingosine-1-phosphate (S1P). S1P is synthesized by phosphorylation of sphingosine (Sph) by two sphingosine kinases, SphK1 and SphK2. They have different subcellular localization and generate distinct pools of S1P with different, and at times opposite, functions [23]. SphK2 is localized inside the nucleus and generates S1P that regulates gene expression [24]. SphK1 resides in the cytosol, and the S1P produced by it can function as a second messenger or be secreted and act as an extracellular ligand, binding to one of the five S1P receptors, S1PR1–5 [25]. These receptors belong to the family of G-protein coupled receptors, and downstream signaling through them regulates multiple secondary signaling molecules, including PI3K, protein kinase-C, phospholipase C, and intracellular calcium [26,27]. The SphK1–S1P axis has been shown to induce proliferation, migration, and differentiation, and is responsible for more-complex cellular responses such as inflammation, immune cell trafficking, and vascular development [28,29,30].

It has been shown that plasma contains high levels of S1P [31,32,33] and that plasma S1P is an important player in regulating functions of the vascular system ranging from establishment of blood vessels, maturation and homeostasis of the vascular system, and blood flow [33,34]. S1P signaling through S1PR1 is necessary during development for blood vessels to properly mature. A global knockout of *S1p1* is embryonic lethal between E12.5 and E14.5 due to the inability of mural cells/pericytes to be recruited and migrate to new vessels. This leads to incomplete maturation of the vessel structure and results in excess hemorrhage [35]. Contact between pericytes and endothelial cells is made through S1PR1 signaling, which induces N-cadherin expression [36]. Conditional knockouts of *S1p1* in vascular endothelial cells lead to a loss of vascular endothelial (VE) cadherin expression between vascular endothelial cells, leading to leaky vessels and hyper-sprouting at leakage points, leading to further disruption [37]. The concentration of S1PR1 increases as tension between endothelial cells increases. This ensures that adherens junctions remain stable, and the vessels remain flow-competent [37]. In this way, S1P signaling opposes angiogenic sprouting and results in maturation of the blood vessel [37,38]. Studies in *Sphk1* and *Sphk2* double-knockout mice show that ablation of S1P production results in a similar phenotype as seen with the global *S1p1* knockout mice [34]. However, individual knockout mice for *Sphk1* or *Sphk2* do not have overwhelming gross phenotypes in vascular patterning [34]. Sphingosine-1-phosphate in the blood is largely derived from erythrocytes, and if both *Sphk1* and *Sphk2* are conditionally knocked out in erythrocytes, the result is embryonic lethality between E11.5 and E12.5 [39,40]. In addition, overexpression of SphK1 in endothelial cells cultured in a Matrigel system increases extracellular S1P, leading to an increase in angiogenesis and vascular maturation [39].

S1P has also been shown to play crucial role in tissue fibrosis [41,42], and S1P levels in plasma and/or the concerned tissue are associated with multiple fibrotic factors, including transforming growth factor β (TGF-β) and platelet-derived growth factor (PDGF) [43]. Both intracellular and extracellular S1P have been shown to be involved in tissue fibrosis, and an increase in the plasma level of S1P has been associated with multiple fibrosis-related diseases. Further, the use of antibody treatment neutralizing S1P can effectively reduce TGF-β-mediated collagen production [44]. Previous studies from our group have indicated that S1P is a critical component of corneal fibrosis [45,46,47]. However, the role of the SphK1–S1P signaling axis in corneal NV and fibrosis has not yet been elucidated. Here, by using global *Sphk1* knockout (*Sphk1*^−*/*−^) mice and an alkali-burn corneal injury model, we show that decreasing the available pool of S1P can reduce neovascular invasion into the cornea following corneal injury. These findings can help to delineate the role of S1P and the SphK1–S1P signaling axis in corneal NV and fibrosis, leading to better understanding of these devastating pathologies.

## 2. Materials and Methods

### 2.1. Reagents

Primary antibodies used in this study were Armenian hamster anti-CD31 (Developmental Studies Hybridoma Bank (DSHB) clone # 2H8. It was deposited to the DSHB by Bogen, S.A. (DSHB Hybridoma Product 2H8)); anti-NG2 (a gift from Dr. William B. Stallcup, Sanford–Burham Medical Research Institute, La Jolla, CA, USA); anti-Iba1 (Wako, 019-19741). Secondary antibodies were anti-rat–Alexa Fluor 488 (Code # 712-545-150, Jackson ImmunoResearch, West Grove, PA, USA); anti-Armenian hamster–Alexa Fluor 594 (Code # 127-585-099, Thermo Fisher Scientific, Waltham, MA, USA); anti-rabbit–Alexa Fluor 488 (Cat # A-21441, Thermo Fisher Scientific, Waltham, MA, USA); anti-guinea pig–Alexa Fluor 488 (Code # 706-545-148, Thermo Fisher Scientific, Waltham, MA, USA).

### 2.2. Animal Care

All animals utilized in this study were born and raised either at the Dean A. McGee Eye Institute, University of Oklahoma Health Sciences Center (OUHSC) or in the University of Tennessee Health Science Center (UTHSC) vivarium following their respective guidelines for animal housing. All procedures were performed according to the Association for Research in Vision and Ophthalmology Statement for the Use of Animals in Ophthalmic and Vision Research and the OUHSC Guidelines for Animals in Research, and were reviewed and approved by the UTHSC and OUHSC Institutional Animal Care and Use Committees (IACUCs). The mice were maintained from birth under dim cyclic light (5–10 lux, 12 h on/off). *Sphk1*^−*/*−^ mice were a gift from Dr. Richard L. Proia (NIDDK, Bethesda, MD). Approximately 2.5- to 4-month-old *Sphk1*^−*/*−^ and WT littermates were used in this study. Each study group was equally divided among male and female mice.

### 2.3. Blood/Plasma Collection

Mice were euthanized individually by carbon dioxide asphyxiation, and blood was immediately collected from a cardiac puncture into an EDTA-coated syringe. The whole blood was very slowly transferred to an ice cold micro-centrifuge tube coated in EDTA and spun down for 20 min at 2000 rpm at 4 °C. Plasma supernatant was removed and flash-frozen in liquid nitrogen. For complete blood count analysis, the blood in an EDTA microcentrifuge tube was given to the OUHSC Comparative Medicine Department.

### 2.4. Sphingolipid Analysis

Animals were euthanized by carbon dioxide asphyxiation, and the eyes were enucleated. Corneas were dissected away from the globe at the limbal line, flash-frozen in liquid nitrogen, and stored in a −80 °C freezer until further analysis. Sphingolipid analysis was performed using LC-MS at Virginia Commonwealth University following previously published protocol [48,49].

### 2.5. Cornea Alkali Burn

The mice were anesthetized with an intraperitoneal injection of ketamine (100 mg/kg body weight) and xylazine (5 mg/kg body weight). A topical application of 0.5% proparacaine hydroxide (Alcon Laboratories, Fort Worth, TX, USA) was applied to the corneal surface of the eye. A 2 mm round piece of Whatman No. 1 filter paper soaked in 1N NaOH was then applied to the central cornea of the left eye for 20 s under a dissection microscope. One eye was burned in each animal, and the contralateral eye served as a non-burned control. Following the burn procedure, the eye was immediately rinsed with 0.9% sterile saline solution for 20–30 s. A topical antibiotic, Erythromycin Ophthalmic Ointment 0.5% (Bausch & Lomb, Rochester, NY, USA), was then applied to the burned cornea, and the mice were monitored to ascertain that they awoke from the anesthesia. The corneas were harvested as described above on different post-burn days (PBD) and used for immunohistochemistry and quantification for neovascular invasion as described previously [46].

### 2.6. Ethynyl Deoxy Uridine (EdU) Pulse

Following the corneal alkali burn procedure, the mice were returned to housing. Ethynyl deoxy uridine (EdU) injections were done 5 days post burn in a sterile hood. Ethynyl deoxy uridine (Thermo Fisher Scientific, Waltham, MA, USA) was dissolved in sterile 100% DMSO to a working concentration of 2.5 mg/mL. Mice were administered 100 μL of the working stock of EdU via an intraperitoneal injection. Mice were euthanized 7 days post burn, and the eyes were enucleated immediately and fixed in 4% paraformaldehyde (PFA) for one hour. Corneas were dissected from the globe of the eye and permeabilized in 1% Triton X-100 in PBS overnight at 4 °C. Corneas were then blocked in 10% horse serum in 1% Triton X-100 in PBS for one hour at room temperature. The tissue was then washed in 0.1% Triton X-100 in PBS two times for 15 min. The Click-iT reactions were set up using a Click-iT Kit (Thermo Fisher Scientific, Waltham, MA, USA) following manufacturer instructions. After the reaction and staining, stacked images of the neovascularizing cornea were acquired at 3 different locations using an Olympus FV1200 confocal microscope and converted into maximum-intensity projections in FIJI. Each stack was quantified for the total number of tip cells and the number of tip cells positive for EdU. The three stacks per cornea were combined and averaged for each animal for an n equal to one.

### 2.7. Immunohistochemistry

After euthanization, the mice eyes were enucleated and submerged in 4% PFA. After 30 min, the eyes were dissected, removing the cornea from the globe below the limbus. The corneas were allowed to fix for another 15 min then moved into 1% Triton X-100 in PBS to permeabilize overnight at 4 °C. Corneas were blocked in 10% horse serum in 1% Triton X-100 in PBS for one hour and incubated overnight at 4 °C with respective primary antibodies as described in detail in Section 2.1. The following day, they were washed with 0.1% Triton X-100 in PBS and incubated with secondary antibodies for 2 hrs. at room temperature. The corneas were washed with 0.1% Triton X-100 in PBS, stained with DAPI, and flat-mounted on a slide in 50% glycerol mounting media. The corneas were imaged using an Olympus MVX or Olympus FV1200 confocal microscope.

### 2.8. Multiplex Assay

Unburned and alkali-burned corneas were harvested at different post-burn days as mentioned in the Section 3 and were snap-frozen in liquid nitrogen. Total cellular proteins were isolated using tissue protein extraction (T-Per) reagent (Thermo Fisher Scientific, Waltham, MA, USA) with protease inhibitors (Roche, Basel, Switzerland). Four slits were made in each cornea, and they were then homogenized on ice. The protocol for the multiplex setup was followed according to the manufacturer, RD-Systems. This kit was custom-made by RD-Systems with mouse Luminex magnetic beads for the following cytokines and growth factors: Angiopoietin-2, Chemokine (C-C motif) ligand 3 (CCL-3), CCL2/Monocyte Chemoattractant Protein-1 (MCP-1), CCL3/Macrophage Inflammatory Protein-1α (MIP-1α), CCL4/MIP-1β, C-X-C Motif Chemokine-10 (CXCL-10), Fibroblast Growth Factor-basic (FGF-basic), Intercellular Adhesion Molecule-1 (ICAM-1), Interferon-gamma (IFN-γ), Interleukin- 1α (IL-1α), IL-1β, IL-4, IL-6, IL-10, IL-12, IL-17A, Matrix Metalloproteinase-9 (MMP-9), Tumor Necrosis Factor- alpha (TNFα), and Vascular Endothelial Growth Factor (VEGF). The assay was read using a Bio-Rad Bio-Plex 200 System.

### 2.9. Statistics

All statistics were calculated using GraphPad Prism Windows (GraphPad Software, La Jolla, CA, USA). Outlying data points were evaluated by a ROUT test (Q = 1%) and removed from the dataset if they tested positive for being an outlier. Column data were assessed for normality, and t-tests were done for normal data while Mann–Whitney tests were done for non-normal data when comparing two variables. Grouped data were evaluated by two-way ANOVA followed by Bonferroni’s correction for multiple comparisons. A p-value <0.05 was considered significant.

## 3. Results

### 3.1. Sphingolipid Profile Is Altered in the Cornea and Plasma of Sphk1^−/−^ Mice

To determine the expression levels of *Sphk1* and *Sphk2* genes, RNA from both WT and *Sphk1*^−*/*−^ mice corneas (*n* = 4) were subjected to quantitative RT-PCR. Whereas *Sphk1* mRNA was undetectable in *Sphk1*^−*/*−^ mice corneas, expression of *Sphk2* mRNA was significantly higher in the *Sphk1*^−*/*−^ mice as compared to the WT mice corneas. Further, the copy numbers of *Sphk1* mRNA were considerably less than those of *Sphk2* mRNA in the WT corneas (Appendix A). To understand how corneal sphingolipid homeostasis was affected by ablation of the *Sphk1* gene, we analyzed the sphingolipid profiles of the corneas of WT and *Sphk1*^−*/*−^ mice (*n* = 6) by LC/MS. We found that in corneal tissue, the concentrations of S1P and dihydroS1P (dhS1P) were below the threshold of measurement (<0.5 pmol/mg wet-tissue weight). Sphingosine (Sph) and dihydroSph (dhSph) were found in higher concentrations than S1P and knocking out *Sphk1* had no significant effect (Figure 1A). However, we observed that total levels of ceramide, monohexosyl ceramide (HexCer), and SM were significantly higher in the *Sphk1*^−*/*−^ mouse corneas (Figure 1B).

Analysis of individual ceramide species revealed significant increase in 16:0 and significant reduction in 24:1 in the *Sphk1*^−*/*−^ corneas (Figure 1C). Among the HexCer species, we observed significant elevation in the levels of 18:1, 22:0, 24:1, and 24:0, and in 18:0 and 24:1 levels of SM (Figure 1D,E). In addition, we evaluated the overall composition of the individual species by calculating sphingolipids as a mole percentage, and found that there were no significant changes in ceramides, HexCer, or sphingomyelin composition between WT and *Sphk1*^−*/*−^ corneas (Appendix A).

Consistent with the previous study [50] we found a significant reduction in S1P and dhS1P levels in the plasma of *Sphk1*^−*/*−^ mice (Figure 1F). Both Sph and dhSph levels were very low in the plasma; however, they were not significantly different between WT and *Sphk1^-/-^* mice (Figure 1F). Analysis of the individual sphingolipid species by mole percentage revealed a significant increase in ceramide 22:0 in the *Sphk1*^−*/*−^ mice plasma as compared to the WT mice (Figure 2A). However, we did not observe any significant differences among the individual species of HexCer and SM (Figure 2B,C).

### 3.2. Sphk1^−/−^ Mice Have Reduced Corneal Neovascularization following Injury

Since S1P signaling plays an important role in maturation and maintenance of blood vessels [33,34], it is conceivable that it might influence neovascularization (NV) following corneal injury. We used an alkali burn model to induce corneal NV, as described in detail in Materials and Methods. This model allowed for reproducible and consistent NV from all areas of the limbus into the cornea. The corneas of WT and *Sphk1*^−*/*−^ mice were evaluated for invading blood vessels after 10 days of burn injury by probing for CD-31 using immunofluorescence. As depicted in Figure 3A,B, we observed reduced area of invasion (pseudo-colored in yellow) in the *Sphk1*^−*/*−^ corneas as compared to the WT littermates. Using quantitative analysis, we observed a significant reduction to the invasion area in the *Sphk1*^−*/*−^ mice following burn injury (Figure 3C).

We observed that compared to the WT mice, the leading fronts of NV invasion were made up of rounded loops of vessels in the *Sphk1*^−*/*−^ mice (Figure 3B). In the WT mice, the leading fronts had a more jagged appearance at the edge of the vessels pointing to the center of the cornea (Figure 3A). Since sprouting angiogenesis is led by the endothelial tip cells [51], we wanted to know if the tip cells or proliferating tip cells were affected in the *Sphk1^-/-^* animals. To test this, we pulse–chased the alkali burned animals with a thymidine analogue 5-Ethynyl-2′-deoxyuridine (EdU). EdU can become incorporated into the DNA and is used for labelling dividing cells. As described in Material and Methods, both WT and *Sphk1*^−*/*−^ mice were injected with EdU 5 days post burn, and corneas were harvested 2 days after injection. Both unburned and burned corneas were immunoprobed for CD31 to visualize the endothelial cells in addition to EdU and were imaged using an Olympus FV1200 confocal microscope (Figure 4A–D). Quantitation showed that both the total and the EdU-positive tip cells were reduced in number in the KO mice corneas at 7 PBD (Figure 4E,F). Even though the reductions in the *Sphk1*^−*/*−^ corneas were not statistically significant, the lower number of endothelial tip cells in the *Sphk1* KO mice might play a role in protecting these corneas from NV.

Another possible reason for reduced NV in *Sphk1*^−*/*−^ mice could be due to the loss of migration of mural cells in an environment where the level of S1P is reduced. Pericytes migrate to new blood vessels to mature and stabilize the structure [52,53,54]. To observe the migration of pericytes in the newly forming blood vessels in WT and *Sphk1*^−*/*−^ mice, we immunoprobed the alkali-burned corneas by neural/glial antigen 2 (NG2). Our results suggest that, following alkali burn, pericytes effectively line the walls of the newly forming vessels in both WT and *Sphk1*^−*/*−^ mice corneas (Appendix A).

Macrophages travelled through the bloodstream and were recruited to the site of injury [55]. They leave the bloodstream by adhering to and migrating through the blood vessel, making the vessels leaky at their exit sites, which become potential sites for angiogenesis [55,56,57]. We immuno-stained 7-day-post-burn corneas from WT and *Sphk1^-/-^* mice with CD31 to visualize the vessels and IBA1, a marker for macrophage. We observed no significant difference between macrophage egress between the two genotypes. Macrophages can be observed leaving the blood vessels and are present in the surrounding corneal tissue in both WT and *Sphk1*^−*/*−^ mice (Appendix A). We also observed that macrophages migrate to the center of the cornea towards the burn wound, where they can be seen in high numbers in both WT and *Sphk1*^−*/*−^ mice (data not shown). Since it is known that T-cells exit the lymph nodes following an S1P gradient [58,59], we wanted to know if, in the *Sphk1*^−*/*−^ animals, there were different populations of immune cells that may contribute to a reduced-NV phenotype. We collected blood from mice and analyzed the complete blood counts to see if these cells had altered populations. In support of the previous literature [50], we found that platelets, white blood cells, neutrophils, monocytes, and lymphocytes remained unchanged in the blood of KO mice (Appendix A).

Taken together, our data suggest that SphK1 promotes corneal NV after an injury, and ablation of the *Sphk1* gene can reduce corneal NV. This reduction in neo-vascular invasion might be due to a reduced number of endothelial tip cells in the *Sphk1*^−*/*−^ mice corneas. However, pericyte migration or macrophage egress from blood vessels does not play a significant role in reduced NV. Further studies will be necessary to understand the possible mechanism(s) of corneal NV after injury and the role played by the *Spkh1* gene.

### 3.3. Alkali Burn Alters Angiogenic and Inflammatory Cytokine Markers in the Cornea

To understand the effect of alkali burn on cytokine and angiogenic signaling in the cornea of WT and *Sphk1*^−*/*−^ mice, we probed for the levels of angiogenic, fibrotic and inflammatory cytokine markers in non-burned and 1-, 3-, and 7-day post burn (PBD) corneas of both WT and *Sphk1*^−*/*−^ mice using a Luminex assay system from R&D biosystems. We observed an increase in VEGF levels in the burned corneas at 1 PBD compared to unburned corneas of both WT and *Sphk1*^−*/*−^ mice, which reduced to low levels by 3 PBD in both the genotypes. However, we did not observe any significant difference between WT and *Sphk1*^−*/*−^ mice (Figure 5A). The levels of Angiopoetin-2, ICAM-1, and FGF were all increased at 1 PBD and stayed high throughout the experimental time frame (7 PBD) in both WT and *Sphk1*^−*/*−^ corneas compared to the unburned corneas. However, similar to VEGF, we did not observe any significant difference in any of their levels between WT and *Sphk1*^−*/*−^ mice (Figure 5B–D).

Among the inflammatory cytokines, we observed an increase in CCL-3/MIP-1α levels in 1 PBD, which is reduced at 3 PBD in both genotypes. However, the increase in *Sphk1*^−*/*−^ is significantly higher than the WT corneas at 1 PBD (Figure 6A). Similarly, we observed a significant increase in IL-1α in the *Sphk1*^−*/*−^ corneas in 1 PBD as compared to WT corneas, which returns to the same level as that of unburned corneas at 3 PBD. Interestingly, in WT corneas, the level does not change as a result of the injury (Figure 6C). Both MMP-9 and IL-6 levels were increased in the burned corneas at 1 PBD and remained high throughout 7 PBD in both WT and *Sphk1*^−*/*−^ without any significant difference between the genotypes (Figure 6B,D). We did not observe any significant difference between IL-12 levels between unburned and burned corneas of both the genotypes (Figure 6E). As expected after an injury, we saw a reduction in the level of the anti-inflammatory cytokine IL-10 in the burned corneas at 1 PBD that stays low over the experimental course without any significant difference between WT and *Sphk1*^−*/*−^ mice (Figure 6F).

Overall, our data suggest that there is an increased inflammatory response in the burned corneas, which is expected after an injury. However, even though there are some differences between *Sphk1*^−*/*−^ and WT mice immediately after the injury (1 PBD), their angiogenic and inflammatory responses become very similar as the wound-healing process progresses through 7 PBD.

## 4. Discussion

In this study, we characterized the sphingolipid profiles in the corneas of WT and *Sphk1*^−*/*−^ global KO mice for the first time. Only *Sphk*^−*/*−*-*^ global KO mice were used in this experiment because *Sphk1* is the dominantly expressed isoform among the two isoforms (*Sphk1 and Sphk2)* that have been reported in the cornea during inflammation [60]. Previous reports have also shown that *Sphk1* is the primary isoform responsible for S1P that is secreted from cells [61]. Concentrations of S1P and dhS1P were very low in both WT and *Sphk1^-/-^* mice corneas (Figure 1A). This was an expected result since levels of these sphingolipids are consistently reported to be low in tissues [50]. The cornea is also not a lipid-rich tissue and contains low cell density compared to many other tissues. We found that in KO mice, total ceramide content was significantly increased. Individual ceramide species showed no significant differences, except 16:0, which was increased, and 24:1, which was decreased (Figure 1B,C). The composition of ceramides, calculated as total mole percentages, showed little change between WT and KO mice (Appendix A). Interestingly, even though there was a decrease in the ceramide 24:1 level in the *Sphk1^-/-^* mice, there was a 3% increase in the mole percent of that species, suggesting that some compensation was occurring in the overall composition of ceramide species (Appendix A). Ceramide concentrations are normally increased if the salvage pathway for ceramides is blocked by hindering the production of S1P, which might happen due to ablation of the *Sphk1* gene. Levels of HexCer and SM were also increased (Figure 1B). This is most likely in compensation for an increase in ceramides, which the tissue must regulate for the risk of becoming apoptotic. However, it is difficult to explain the significance of changes in species such as Cer C16:0, which increased in the KO cornea, and Cer 24:1, which decreased without a change in the mole% of C24:1. It is also noticeable that C24:1 (lignoceric acid, a bioactive sphingolipid) is one of the major species of sphingolipids in the cornea; along with its decrease in the ceramide group, it is significantly increased in HexCer and SM. This could result from a metabolic shift in the KO cornea that generated more sphingolipids than the controls as a result of more de novo biosynthesis. Sph and S1P are known intracellular inhibitors of ceramide synthases (CerS) [62]. There are six CerS in mammalian cells, which are somewhat specific for the chain length of ceramides they generate [62]. It is not known which CerSs are expressed in corneal cells. The most common are CerS2 and CerS4, which can synthesize long-chain Cer such as C22–C26 [62]. It is known that S1P can allosterically inhibit CerS2 function to regulate Cer–S1P balance in a cell [63]. It, therefore, could be speculated that SphK1-generated S1P is important to keep the Cer concentration in check, and ablation of SphK1 increased the ceramide levels due to lack of this inhibition (Figure 1C–E). We also characterized the blood plasma concentrations of sphingolipids. Consistent with previous research [50] that has characterized *Sphk1^-/-^* mouse serum levels of sphingolipids, we found that S1P and dhS1P levels were significantly reduced in the plasma (Figure 1F). Whereas there is an abundant level of S1P in the plasma, the levels of S1P in the corneas of both genotypes were very low (<1 pmol/mg of tissue). A reduction in the plasma levels of S1P indicates that Sphk1 does not have an exclusive role in maintaining the plasma levels of S1P, and Sphk2 might also contribute to the plasma levels of S1P. The expression of Sphk2 is known to increase as a result of SphK1 loss, and it is also known to functionally compensate the loss of Sphk1 [34,64,65,66]. Similarly, we also observed a significant increase in the mRNA levels of Sphk2 in *Sphk1^-/-^* mice corneas as compared to the WT (Appendix A). The presence of similar but low levels of S1P in the corneas between WT and *Sphk1*^−*/*−^ mice might be due to a higher expression level of Sphk2 observed in the cornea that is responsible for maintaining the critical cellular levels of S1P observed in the cornea. The mole percentage of plasma ceramide showed little change, except for 22:0, which was significantly higher in KO corneas (Figure 2A). The reduction in S1P production might attribute to the increase in ceramide level. All other sphingolipid species remained unchanged (Figure 1D,E and Figure 2).

We show here for the first time that NV in the cornea can be reduced by knocking out the *Sphk1* gene (Figure 3A,B). Studies using tumor models have also shown some reduction in angiogenesis by pharmaceutically targeting SphK1, but these studies contradict one another and are further complicated by the role that S1P plays in tumor cell growth and proliferation [35,67,68,69]. The S1PR1 receptor is critical for blood vessel development and maturation, and *S1p1* knockout mice are embryonic lethal due to hemorrhage of developing vasculature [35]. In conditional *S1p1* knockout mice, in which vascular endothelial cells are targeted, there is a breakdown of vessel integrity and hyper-sprouting at the angiogenic front [37]. The source of S1P for S1PR1 on vascular endothelial cells comes from the serum/plasma [39,70]. Here, we show that a reduction in plasma S1P concentrations correlates with reduced NV in the cornea following corneal injury in form of alkali burn. We also observed that in KO mice, the leading edge of NV tended to be more rounded off in comparison to the more jagged appearance seen in WT corneas. To investigate this phenotype, we did a pulse–chase experiment with EdU to see if proliferation at the leading edge of NV was altered. We observed that, albeit statistically non-significant, the numbers of both tip cells and EdU-positive tip cells were reduced in the *Sphk1*^−*/*−^ corneas (Figure 4). The reduced number of tip cells in *Sphk1*^−*/*−^ mice might play a role in reducing NV in those corneas. This was a surprising result, because based on the literature, we expect to see more sprouting to occur in KO corneas. Decreased S1P signaling in angiogenesis results in hyper-sprouting and no formation of an endothelial tube [37]. It is also well-established that S1P signaling through S1PR1 blocks angiogenic sprouting, as VE- and N-cadherins are produced by the vascular endothelium, triggering maturation of the vessel [38,71]. Increased sprouting in the cornea may not occur because the tissue itself is programmed to block angiogenic sprouting. The cornea actively reduces VEGF-A levels in the tissue by producing VEGF traps [3].

It is possible that the reduction in NV we observed is due to reduced ability of vascular endothelial cells to mature into a blood-flow competent structure because of decreased S1P signaling, while increased sprouting is constantly blocked by the corneal tissue. We also cannot rule out that an increase of ceramide could play a role in blocking the formation of mature vessels. Ceramides have been shown to affect established vasculature, primarily by affecting vascular tone and the ability of the cells to sense oxygen [72,73]. Increased ceramides lead to hardening of arteries and decreased vasodilation ability [74]. This, in part, is due to the ability of ceramides to induce calcification of vascular smooth muscle cells [75]. The effect of ceramides in angiogenesis and NV is not well understood. However, it has been shown that migration and maturation of pulmonary artery smooth muscle cells depends on ceramide and S1P production [76]. Further studies using the alkali-burn model in mice with overexpression of ceramide synthase genes would be helpful in showing how ceramides affect angiogenesis and neovascularization.

We also investigated if a decrease in plasma levels of S1P would alter pericyte migration to the newly forming blood vessels in the cornea. We found pericytes were encasing blood vessels in both WT and *Sphk1*^−*/*−^ mice after an alkali burn (Appendix A). While this indicates that the blood vessels present in the KO corneas have some capacity to mature properly, it does not fully show that pericyte migration is normal. In vivo studies using florescent protein-expressing pericytes would be needed to fully test if migration is affected. However, models that show migration issues in pericytes, primarily *Sphk1* and *Sphk2* double-KO mice and *S1p1* KO mice, show that no pericytes attach to the developing endothelial tube [35,37]. Since we do observe pericytes attaching to the blood vessels in both genotypes, it is unlikely that pericyte migration is a major contributing factor in the NV reduction we observe. Corneas from WT and KO mice were also stained for macrophages to see if they were recruited and then egress from blood vessels, but no difference was observed between WT and *Sphk1* KO mice (Appendix A).

We also analyzed cytokines and angiogenic factors in these mice (Figure 5 and Figure 6). We found increased VEGF at 1 PBD in the burned corneas of both genotypes. Interestingly, we found that in the KO corneas, there is a trend towards increased expression of VEGF (Figure 5A). Levels of ICAM-1 increase from 1 PBD through 7 PBD, indicating increased vasculature, similar in both WT and KO mice (Figure 5C). A similar trend was also observed in Angiopoetin-2 (Ang2) and FGF levels (Figure 5B,D). The role of Ang2 in angiogenesis is context-dependent, and it can be both pro- and anti-angiogenic [77]. Ang2 expression is known to be induced by inflammation and has been reported to induce permeability and angiogenesis in the presence of VEGF [77]. An increasing trend of both VEGF and Ang2 at 1 PBD indicates that their interplay might have a role in corneal NV. Further studies will be needed to properly understand their role in the context of corneal injury.

The first response after any tissue injury is inflammation mediated by inflammatory cytokines and chemokines, secreted primarily by the innate immune cells [78]. We followed the expression levels of markers that might play crucial roles in inflammation after injury and for neovascularization to understand the molecular pathophysiology in corneal injury and neovascularization. The pro-inflammatory cytokines IL-1α and IL-6 are known to be secreted by macrophages, neutrophils, and endothelial cells, while anti-inflammatory IL-10 is primarily secreted by monocytes [79,80,81]. All these cells are known to be involved in tissue injury and neovascularization. In addition, MMP9 is pro-angiogenic, while IL-12 is known to play an anti-angiogenic role; therefore, they might be important contributors in neovascularization following corneal injury [82,83,84]. As expected after an injury, we observed a decrease in the level of anti-inflammatory cytokine IL-10 in the burned corneas of both genotypes (Figure 6F). Levels of CCL-3 and IL-1α, both indicative of increased inflammation and macrophage activation, were significantly increased in KO corneas (Figure 6A,C). It is known that increased levels of IL-1α can induce expression of CCL-3, and in doing so, activate and increase migration of macrophages to the site of injury [85]. Macrophages are known to take up ceramides and become activated, which in turn upregulates ceramide synthesis [86], leading to increased phagocytosis and apoptosis [87]. Interestingly, in cultured macrophages in which SphK1 was knocked down, increased apoptosis was observed after the addition of TNFα and lipopolysaccharide [87]. However, it has been shown that murine bone marrow-derived macrophages do not require SphK1 or SphK2 to trigger activation or cytokine inflammation response [88]. The increase in CCL-3 and other inflammatory markers can also be attributed to increased levels of ceramide in the *Sphk1^-/-^* corneas. The levels of CCL-3 and IL-1α were both significantly increased in the KO corneas only on PBD 1; the levels were significantly reduced by PBD 3 and stayed low throughout the experimental period. It could be possible that the level of ceramide, which is already significantly higher in KO corneas (Figure 1B), is further increased in the acute phase of corneal injury and contributes to the higher levels of CCL-3 and IL-1α. Detailed lipidomic analysis of the burned corneas will be needed to fully understand the connection between ceramide, S1P, and the pro-inflammatory cytokines. However, in our model, even with an increase in inflammation markers and level of ceramides, KO corneas show significantly reduced NV.

This study, for the first time, compared the sphingolipids in WT and *Sphk1^-/-^* mice corneas. It also established a role of the SphK1–S1P axis in corneal neovascularization after an injury. S1P has multiple roles, including regulation of inflammation through immune cell trafficking, both pro- and anti-apoptotic roles in a context- and tissue-dependent fashion, and regulation of vascular integrity and normal angiogenesis, underscoring the complexity of S1P metabolism and signaling. Even though the molecular mechanism is still unclear and all the players yet to be identified, our work underscores the role bioactive sphingolipids play in maintaining corneal transparency. Further studies of preclinical corneal injury models will help delineate how the balance between the levels of ceramide and S1P in the cornea dictates the outcome of corneal wound healing and neovascularization.

## Figures and Tables

**Figure 1 cells-11-02914-f001:**
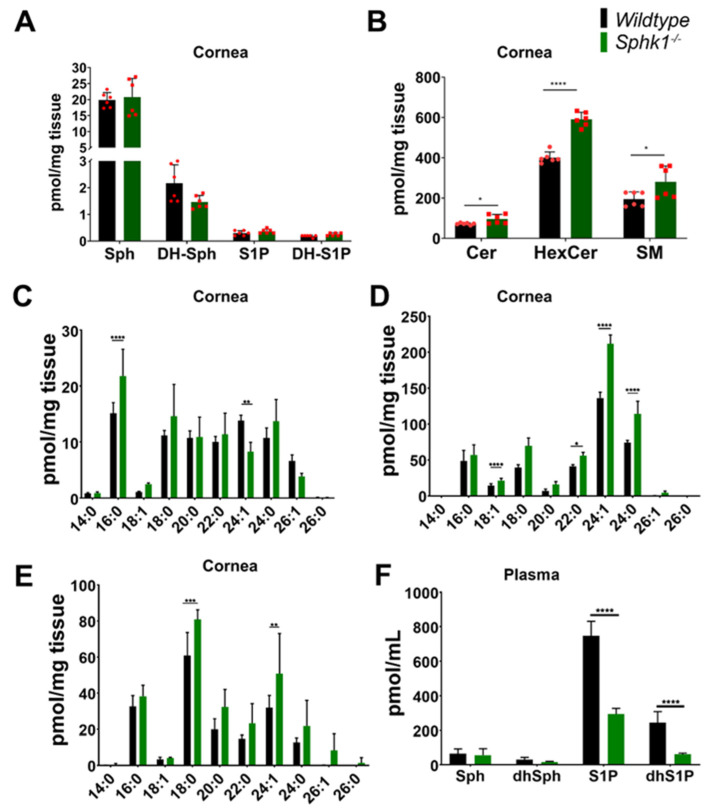
Ablation of *Sphk1* alters sphingolipid homeostasis in the cornea and plasma. Analysis of sphingolipids from WT and *Sphk1*^−*/*−^ mice corneas (*n* = 6) showing total sphingolipid levels (**A**,**B**) with individual species of (**C**) ceramide, (**D**) HexCer, and (**E**) sphingomyelin. Total sphingolipid levels in WT and *Sphk1*^−*/*−^ mice plasma are depicted in (**F**). (Values are mean ± SD, * *p* ≤ 0.05, ** *p* ≤ 0.01, *** *p* ≤ 0.001, **** *p* ≤ 0.0001).

**Figure 2 cells-11-02914-f002:**
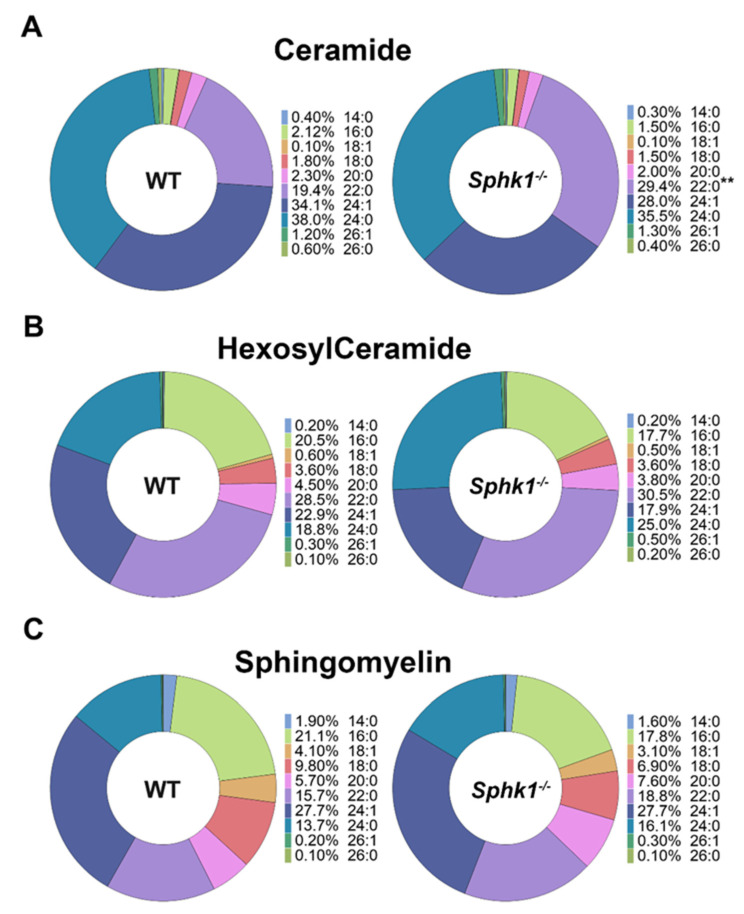
Plasma sphingolipids were altered in *Sphk1*^−*/*−^ mice. Sphingolipid analysis of WT and *Sphk1*^−*/*−^ mice plasma (*n* = 5) showing mole percent composition of (**A**) ceramide, (**B**) HexCer, and (**C**) sphingomyelin. (Values are mean ± SD, ** *p* ≤ 0.001).

**Figure 3 cells-11-02914-f003:**
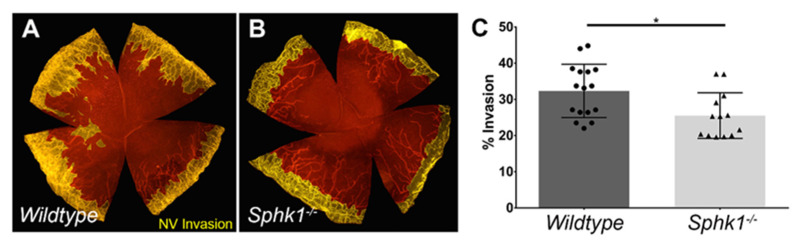
Ablation of *Sphk1* reduce corneal neovascularization following alkali burn. Mice corneas were stained ten days post alkali burn with CD31 (**red**) to visualize vasculature. Representative image in pseudo-color (**yellow**) showing neovascular invasion in (**A**) WT (*n* = 16) and (**B**) *Sphk1*^−*/*−^ (*n* = 15). (**C**) Quantitation of percent neovascular invasion into the cornea. (Values are mean ± SD, * *p* ≤ 0.05).

**Figure 4 cells-11-02914-f004:**
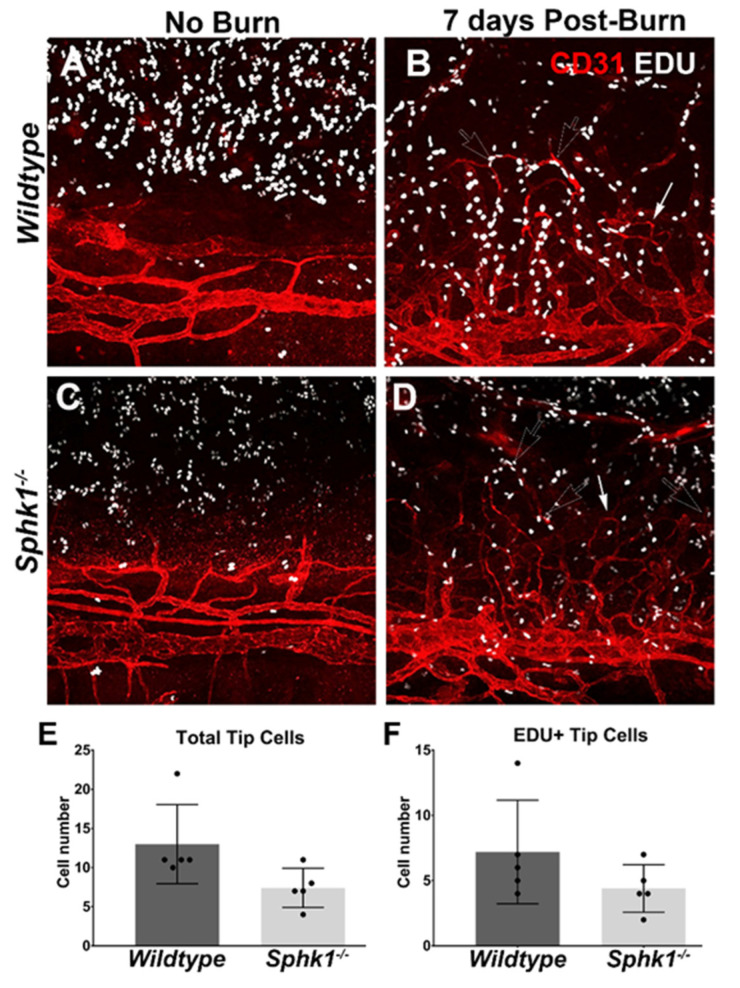
Ablation of *Sphk1* does not affect the numbers of total and proliferating tip cells following alkali burn. WT and *Sphk1*^−*/*−^ mice corneas (*n* = 5) were subjected to alkali burn. The mice were pulsed with EdU 5 days post burn (PBD), and corneas were collected on 7 PBD. Representative image showing WT (**A**) no-burn and (**B**) 7 PBD, and *Sphk1*^−*/*−^ (**C**) no-burn and (**D**) 7 PBD. Blood vessels are stained for CD31 (red), and proliferating cells are stained for EdU (pseudo-white). White-outlined arrowhead points to EdU-positive proliferating tip cells, and solid white arrowhead points to non-proliferating tip cells at the time of the EdU pulse. Quantitation of 7 PBD corneas showing (**E**) total tip cells and (**F**) EDU-positive tip cells. (Values are mean ± SD).

**Figure 5 cells-11-02914-f005:**
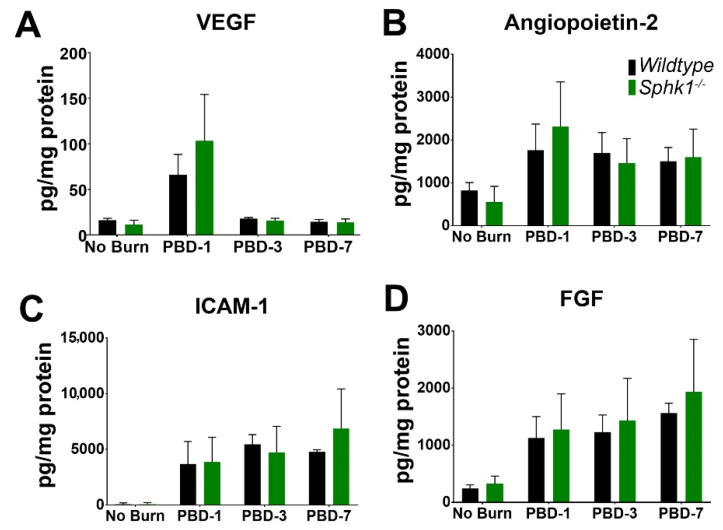
Angiogenic and fibrotic markers are altered in the cornea following alkali burn. WT and *Sphk1*^−*/*−^ mice corneas (*n* = 3) were harvested 1, 3, and 7 days post alkali burn, along with unburned controls, and the levels of angiogenic and fibrotic markers VEGF (**A**), Angiopoitin-2 (**B**), ICAM-1 (**C**) and FGF (**D**) were measured by Luminex assay (Values are mean ± SD).

**Figure 6 cells-11-02914-f006:**
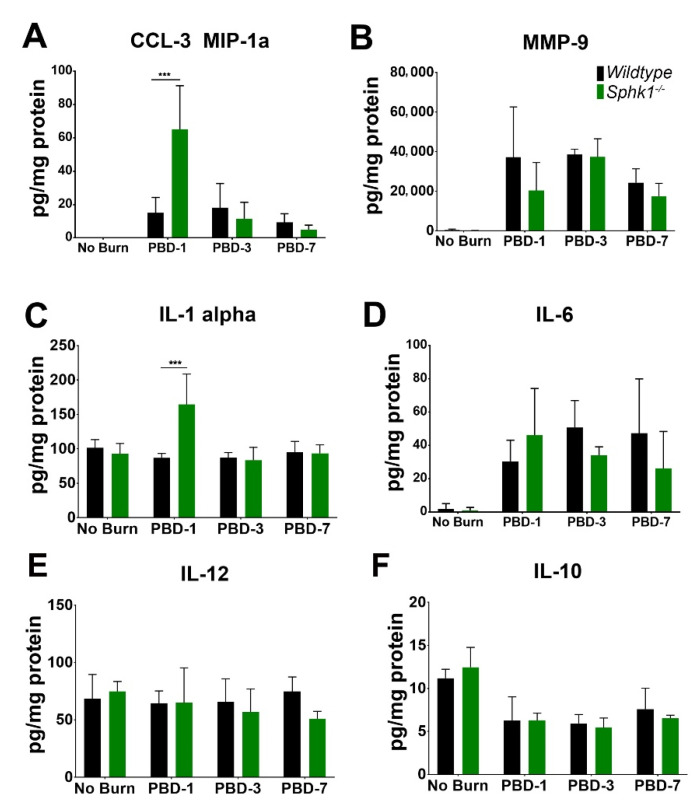
Inflammatory markers are altered in the cornea following alkali burn. WT and *Sphk1^-/-^* mice corneas (*n* = 3) were harvested 1, 3, and 7 days post alkali burn together with unburned controls, and the levels of inflammatory cytokines CCl-3 (**A**), MMP-9 (**B**) IL-1α (**C**), IL-6 (**D**), IL-12 (**E**) and IL-10 (**F**) were measured by Luminex assay. (Values are mean ± SD, *** *p* ≤ 0.05).

## Data Availability

Not applicable.

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
