# Peer review of "Ablation of Sphingosine Kinase 1 Protects Cornea from Neovascularization in a Mouse Corneal Injury Model"

_cells, 2022, doi:10.3390/cells11182914_

Round 1

Reviewer 1 Report

In this study the authors examine the role of sphingosine kinase 1 (Sphk1) in corneal neovascularization after an alkali burn using a mouse corneal alkali burn model.

The authors provide a very nice introduction of S1P and the enzymes Sphk1 and Sphk2 that phosphorylate Sphingosine. The study shows that the area of neovascularization in Sphk1-/- mice was reduced. The KO mouse corneas showed lower endothelial tip cells that contribute to vascular sprouting. They show that the KO corneas have fewer proliferating endothelial tip cells compared to WT. They further ruled out any differences in migrating pericytes or macrophages in the NV area. Of all the inflammatory markers tested, only MIP1 a and IL- 1 alpha were increased in the KO corneas after injury.

This study is generally well done with suitable numbers of mice per trial. The following are concerns that require some clarifications.

1.     What was the justification for selecting IL-1 alpha? Why not IL-1 beta as release of this cleaved and activated cytokine is linked to inflammosome signaling inflammatory caspases? A better justification of why the authors selected to measure IL-1 alpha, MMP9, IL-6, IL-12 and IL-10 is needed. If they are looking for differences in Th1 and Th2 cytokines this could have been done using preset multiplex bead arrays.

2.     The authors see reduction of S1P in the plasma of the KO mice yet the blood cell count shows no difference in circulating lymphocytes. Monocyte levels may be lower but this was not significant. What time point after injury was the blood analyzed? Or was this all at baseline? Ideally this should be analyzed by flow cytometry on individual animals at baseline and probably at the early time point after injury (when they see differences in pro0-inflammatory cytokines), gating for immune cell subsets.

3.     They observed increased CCL3/MIP1 a levels in the cornea. The authors may need to better discuss the potential regulation of this cytokine by ceramides or something that connects it to S1P or Sphk 1.  Which cells secrete CCL3 and what specific inflammation/injury related signals (ceramides?) induce this and how is it modulated by Sphk1?

4.     If the reduced NV in SphK1 corneas is due to reduced proliferation of endothelial tip cells, the authors could consider some in vitro cell culture assays to dissect underlying signaling mechanism as this is a key finding in the study.

5.     Finally, the authors profiled sphingolipids in the cornea and found an increase in the 16:0 ceramide and a decrease in 24:1 in the Sphk1 -/- mouse corneas. What is the significance of this? This could be better explained in the Discussion.

Other minor comments

The following are suggestions that may improve general presentation of the manuscript.

1)     Fig 1 AB: show individual mouse data points in the bar graphs

2)    Since Fig. 2 is not discussed much why not move this in the supplemental figure section. Also since all panels are plasma… they do not each need a subheading.

3)    Since Figure 1 has both cornea and plasma data, a subheading for the cornea and plasma graphs may be easier for the reader.

4)    Figure 4: The WT panels seem at a slightly higher mag. IS that true or is it that the blood vessels in the WT are more robust looking? Scale bar would help.

5)    The first paragraph in the discussion is a bit confusing.

-       Line 358  should say Sphk1 isoform and not Sphk1-/-

-       Line 361 -  secreted or excreted?

-       Line 362 – should read WT and Sphk1-/- mouse corneas

6) Reference 58 -  please cite the original article from the Susan Schwab lab

Reviewer 2 Report

The manuscript authored by Wilkerson, et al. reports that ablation of sphingosine kinase 1 (SphK1) protects cornea from neovascularization in a corneal injury mouse model. It may have translational implication for future studies. 

Major concerns

1. SphK1 animal model: age and gender need to be provided for both wildtype and SphK1 knockout mice. Authors are encouraged to provide protein expression levels of SphK1 and SphK2 in this manuscript. 

2. Figure 1. The levels of S1P were reduced in plasma of SphK1 knockout mice. Nevertheless, there was no difference in the levels of S1P in cornea tissue between wild type and Sphk1 knockout mice. This tissue-dependent discrepancy needs to be addressed. 

3. According to Fig. 1A, S1P levels in cornea were below 1 pmol/mg tissue. Measurement of SphK1 and SphK2 of cornea in control and disease model will be fundamental to address the role of S1P in this manuscript. 

4. Pharmacological inhibitors (SKI-178 or others) of SphK1 may be used to confirm the results.

Minor concerns

1. The antibodies and company information need to be provided in 2.7.

Round 2

Reviewer 2 Report

The concerns have been addressed appropriately by authors in this revised manuscript.